# The Influence of Adolescent Sport Participation on Body Mass Index Tracking and the Association between Body Mass Index and Self-Esteem over a Three-Year Period

**DOI:** 10.3390/ijerph192315579

**Published:** 2022-11-24

**Authors:** Robert J. Noonan

**Affiliations:** 1Faculty of Health and Wellbeing, University of Bolton, Bolton BL3 5AB, UK; r.noonan@bolton.ac.uk; 2Department of Psychology, University of Liverpool, Bedford Street South, Liverpool L69 7ZA, UK

**Keywords:** sport, self-esteem, body mass index, obesity, adolescents

## Abstract

This study aimed to (1) investigate gender-specific characteristics associated with low sport participation among UK adolescents, and (2) assess gender-specific BMI tracking, and gender-specific associations between BMI and self-esteem based on different levels of adolescent sport participation. Participants were 9046 (4523 female) UK adolescents. At 11- and 14 years self-esteem was self-reported and BMI was calculated from objectively measured height and weight. At 11- years sport participation was parent-reported. Gender-specific sport participation quartile cut-off values categorised boys and girls separately into four graded groups. Gender-specific χ^2^ and independent samples t tests assessed differences in measured variables between the lowest (Q1) and highest (Q4) sport participation quartiles. Adjusted linear regression analyses examined BMI tracking and associations between BMI and self-esteem scores. Gender-specific analyses were conducted separately for sport participation quartiles. Compared to Q4 boys and girls, Q1 boys and girls were more likely to be non-White, low family income, have overweight/obesity at 11 years and report lower self-esteem at 11 years and 14 years. BMI at 11 years was positively associated with BMI at 14 years for boys and girls across sport participation quartiles. BMI at 11 years was inversely associated with self-esteem scores at 11 years for Q1 and Q2 boys, and Q1 and Q4 girls. BMI at 11 years was inversely associated with self-esteem scores at 14 years for Q1, Q3 and Q4 boys, and Q1, Q2, Q3 and Q4 girls. Gender and sport participation influence BMI tracking and the BMI and self-esteem association among adolescents.

## 1. Introduction

Preventing overweight among adolescents (period defined as 10 to 19 years) is among the greatest public health challenges facing the UK and other high-income countries. The global prevalence of overweight among adolescents has increased markedly in the past half century [1]. In the UK, over one-third of 10- and 11-year olds have overweight [2]. There is strong evidence linking childhood overweight with adolescent overweight [3,4], and gender differences in childhood overweight and body mass index (BMI) are well established [5,6]. There is also some evidence to suggest that gender influences the stability of BMI (referred to as tracking herein) during adolescence [7]. However, because studies tend to only control for gender in analyses, further gender stratified analyses are warranted to elucidate potential reasons for gender differences in BMI tracking during adolescence [6].

Self-esteem relates to a person’s evaluation of their own worth [8]. Adolescence is a period of increased vulnerability for low self-esteem [9] which is a key component of various psychopathologies including anxiety and depression in adolescence and adulthood [10,11,12]. Girls tend to report lower self-esteem than boys [13,14], and some [15,16], but not all studies [17] have reported lower self-esteem among overweight compared to healthy weight adolescents. While both genders typically experience a decline in self-esteem during the transition from childhood to adolescence, the decline is thought to be greater in girls than boys [18], and girls with obesity have been found to report lower self-esteem than boys with obesity [19]. Presently missing from the literature is an understanding of whether the longitudinal relationship between BMI and self-esteem is consistent for girls and boys. Studies in this area have tended to be cross-sectional in design or have combined boys and girls in analyses, despite gender differences in BMI and self-esteem [20,21,22].

Sports participation has a positive influence on adolescent self-esteem [23,24,25]. There is also some evidence to suggest that sport participation is inversely associated with overweight [26] and BMI [27], and BMI is inversely associated with self-esteem among adolescents [20,21]. However, evidence is inconclusive in this area [28,29,30]. While gender differences in sport participation levels have been reported [19], currently missing from the literature is an understanding of gender-specific characteristics associated with low sport participation among UK adolescents. As such, the extent of inequalities in UK adolescent sport participation is unknown. This is an important line of enquiry given the international interest in enhancing adolescent sport participation and reducing participation inequalities [31]. Furthermore, an improved understanding of the influence of regular sport participation on BMI tracking and the BMI and self-esteem relationship can contribute to more effective sports programmes and health policies aimed at improving adolescent health and reducing health inequalities. Therefore, to fill these knowledge gaps, the present study aimed to (1) investigate gender-specific characteristics associated with low sport participation among UK adolescents, and (2) assess gender-specific BMI tracking, and gender-specific associations between BMI and self-esteem based on different levels of adolescent sport participation.

## 2. Materials and Methods

### 2.1. Study Design and Participants

Data were from sweep five and six of the UK Millennium Cohort Study (MCS). The MCS is a nationally representative UK sample of children born between September 2000 and January 2002. The sample design allowed for over-representation of ethnic minority and socially disadvantaged families. The first sweep of data collection was conducted between 2001 and 2002, and contained information on 18 819 children in 18 533 families, collected from parents/carers (referred to as parents herein) at home addresses when children were 9 to 11 months old [32]. Subsequent surveys were administered at 3, 5, 7, 11 and 14 years. This study uses data collected on children at 11 and 14 years. There were 9046 singleton children (4523 female) with complete data for the variables of interest at 11 and 14 years. The analysis did not require additional ethical approval. Ethical approval for the 11 (fifth) and 14 years (sixth) sweeps of the MCS were granted by the Northern and Yorkshire multicentre research ethics committee (Ethics Committee reference: 11/YH/0203) and the National Research Ethics Service Research Ethics Committee London-Central (13/LO/1786), respectively.

### 2.2. Measures

#### 2.2.1. Body Mass Index and Weight Status

At 11- and 14 years participants had their height and weight measured objectively and BMI was calculated from height and weight (kg/m^2^). For descriptive purposes, three weight status categories; normal weight (including underweight), overweight and obesity were determined based on the International Obesity Taskforce age- and sex-specific BMI cut-points [33].

#### 2.2.2. Self-Esteem

At 11- and 14 years participants completed a shortened version of Rosenberg’s Self-Esteem Scale [8], comprising five items reflecting a positive view of self. Item responses were reported on a 4-point scale ranging from strongly disagree to strongly agree. Responses were summed to provide a score ranging from 5 to 20, with higher scores indicating greater self-esteem. The alpha coefficient inter-item reliability was 0.74.

#### 2.2.3. Sport Participation

At 11 years parents reported how often per week their child participated in sport clubs or classes (e.g., swimming, gymnastics, football). Responses were recorded on a 7-point Likert scale, ranging from not at all to ≥five days per week. The sport participation measure has been used previously [26]. Gender-specific sport participation quartile cut-off values were calculated and used to categorise participants into four graded groups representing the lowest (Q1) through to the highest level of sport participation (Q4). A dichotomous variable was also created to represent the lowest (Q1) vs. highest (Q4) sport participation quartiles.

#### 2.2.4. Covariates

Covariates were selected based on previous evidence [34,35,36]. At 11 years participant gender (male or female) and ethnicity were parent reported. Ethnicity categories were based on census categories [37] and included White, Mixed, Indian, Pakistani and Bangladeshi, Black or Black British, and Other Ethnic group. Parent reported family income was assessed using quintiles of household income equivalised according to the Organisation for Economic Co-operation and Development household equivalence scale [38].

### 2.3. Analysis

All analyses were conducted using SPSS v. 25 (SPSS Inc.; Chicago, IL, USA) and statistical significance was set at *p* < 0.05. Descriptive statistics (mean ± SD or %) were first calculated for the sample. Given established gender differences in BMI, self-esteem and sport participation levels all analyses were conducted separately for boys and girls. To assess gender-specific characteristics associated with low sport participation among UK adolescents and determine the extent of inequality in UK adolescent sport participation (study aim one) only data for the lowest (Q1) and highest (Q4) sport participation quartiles were used in analyses. For the purposes of study aim one, ethnicity (White vs. non-White), family income (lowest quantile vs. above lowest quantile) and weight status (normal weight vs. overweight/obesity) variables were dichotomised. Gender-specific Q1 and Q4 sport participation quartile differences in dichotomised (i.e., ethnicity, family income and weight status) and continuous variables (i.e., BMI and self-esteem) were then assessed with chi-square tests and independent samples t-tests, respectively. Odds ratios and Cohen’s d values were also calculated as a measure of effect size. A series of adjusted linear regression analyses were conducted using the full sample to assess gender-specific BMI tracking, and gender-specific associations between BMI and self-esteem based on different levels of adolescent sport participation (study aim two). The predictor variable was BMI at 11 years. The three outcome variables were BMI at 14 years and self-esteem scores at 11 years and 14 years. Study aim two analyses were adjusted for ethnicity and family income quantile at 11 years.

## 3. Results

### 3.1. Study Aim One

Sample characteristics are presented in Table 1. Table 2 shows gender-specific Q1 and Q4 sport participation quartile differences in dichotomised (i.e., ethnicity, family income and weight status) and continuous variables (i.e., BMI and self-esteem). Q1 boys were more likely to be non-White ethnicity (OR = 2.34; 95% CI = 1.88 to 2.90; *p* < 0.001), in the lowest family income quantile (OR = 3.71; 95% CI = 2.97 to 4.63; *p* < 0.001), and have overweight/obesity at 11 years (OR = 1.53; 95% CI = 1.27 to 1.83; *p* < 0.001) and 14 years compared to Q4 boys (OR = 1.67; 95% CI = 1.39 to 2.00; *p* < 0.001). These boys also had a higher BMI at 11 years (d = 0.2; *p* < 0.001) and 14 years (d = 0.2; *p* < 0.001), and reported lower mean self-esteem scores at 11 years (d = 0.2; *p* < 0.001) and 14 years (d = 0.3; *p* < 0.001). Compared to Q4 girls, Q1 girls were more likely to be non-White ethnicity (OR = 5.50; 95% CI = 4.34 to 6.97; *p* < 0.001), in the lowest family income quantile (OR = 5.15; 95% CI = 4.10 to 6.48; *p* < 0.001), and have overweight/obesity at 11 years (OR = 1.26; 95% CI = 1.06 to 1.51; *p* = 0.01). Q1 girls also reported lower mean self-esteem scores at 11 years (d = 0.2; *p* < 0.001) and 14 years (d = 0.2; *p* < 0.001).

### 3.2. Study Aim Two

Table 3 presents adjusted regression analyses showing age 11 BMI as a predictor of age 14 BMI, age 11 self-esteem scores and age 14 self-esteem scores for each sport participation quartile by gender.

#### 3.2.1. Age 11 BMI and Age 14 BMI

Adjusted linear regression analyses revealed that BMI at 11 years was consistently positively associated with BMI at 14 years for Q1 boys (B = 1.00; 95% CI = 0.96 to 1.04; *p* < 0.001), Q2 boys (B = 1.00; 95% CI = 0.96 to 1.04; *p* < 0.001), Q3 boys (B = 1.01; 95% CI = 0.97 to 1.05; *p* < 0.001), and Q4 boys (B = 0.87; 95% CI = 0.83 to 0.90; *p* < 0.001), as well as Q1 girls (B = 0.97; 95% CI = 0.93 to 1.00; *p* < 0.001), Q2 girls (B = 0.88; 95% CI = 0.84 to 0.92; *p* < 0.001), Q3 girls (B = 0.98; 95% CI = 0.94 to 1.02; *p* < 0.001), and Q4 girls (B = 0.94; 95% CI = 0.90 to 1.02; *p* < 0.001).

#### 3.2.2. Age 11 BMI and Age 11 Self-Esteem

BMI at 11 years was inversely associated with self-esteem scores at 11 years for Q1 boys (B = −0.06; 95% CI = −0.09 to −0.02; *p* < 0.01), Q2 boys (B = −0.05; 95% CI = −0.09 to −0.01; *p* < 0.05), Q1 girls (B = −0.06; 95% CI = −0.09 to −0.02; *p* < 0.01), and Q4 girls (B = −0.04; 95% CI = −0.07 to −0.02; *p* < 0.05).

#### 3.2.3. Age 11 BMI and Age 14 Self-Esteem

BMI at 11 years was inversely associated with self-esteem scores at 14 years for Q1 boys (B = −0.05; 95% CI = −0.09 to −0.00; *p* < 0.05), Q3 boys (B = −0.05; 95% CI = −0.10 to −0.00; *p* < 0.05), and Q4 boys (B = −0.04; 95% CI = −0.08 to −0.00; *p* < 0.05), as well as Q1 girls (B = −0.10; 95% CI = −0.15 to −0.06; *p* < 0.001), Q2 girls (B = −0.12; 95% CI = −0.16 to −0.07; *p* < 0.001), Q3 girls (B = −0.06; 95% CI = −0.12 to −0.01; *p* < 0.05) and Q4 girls (B = −0.07; 95% CI = −0.12 to −0.03; *p* < 0.01). Associations were consistently greater for girls than boys.

**Table 3 ijerph-19-15579-t003:** Regression analyses showing age 11 BMI as a predictor of age 14 BMI, age 11 self-esteem scores and age 14 self-esteem scores for each sport participation quartile by gender. Adjusted regression coefficients and 95% confidence intervals.

	Age 14 BMI		Age 11 Self-Esteem		Age 14 Self-Esteem	
	B (95% CI)	*p* Value	B (95% CI)	*p* Value	B (95% CI)	*p* Value
Boys						
Q1	1.00 (0.96 to 1.04)	<0.001	−0.06 (−0.09 to −0.02)	0.002	−0.05 (−0.09 to −0.00)	0.04
Q2	1.00 (0.96 to 1.04)	<0.001	−0.05 (−0.09 to −0.01)	0.03	−0.02 (−0.07 to 0.03)	0.45
Q3	1.01 (0.97 to 1.05)	<0.001	−0.02 (−0.06 to 0.02)	0.32	−0.05 (−0.10 to −0.00)	0.04
Q4	0.87 (0.83 to 0.90)	<0.001	−0.00 (−0.03 to 0.03)	0.84	−0.04 (−0.08 to −0.00)	0.03
Girls						
Q1	0.97 (0.93 to 1.00)	<0.001	−0.06 (−0.09 to −0.02)	0.002	−0.10 (−0.15 to −0.06)	<0.001
Q2	0.88 (0.84 to 0.92)	<0.001	−0.01 (−0.04 to 0.02)	0.50	−0.12 (−0.16 to −0.07)	<0.001
Q3	0.98 (0.94 to 1.02)	<0.001	−0.02 (−0.06 to 0.02)	0.37	−0.06 (−0.12 to −0.01)	0.03
Q4	0.94 (0.90 to 0.97)	<0.001	−0.04 (−0.07 to −0.02)	0.02	−0.07 (−0.12 to −0.03)	0.002

BMI, body mass index; CI: confidence interval; Q1 and Q4 represent lowest and highest sport participation quartiles, respectively. Adjusted for ethnicity and family income quantile.

## 4. Discussion

This is the first study to (1) investigate gender-specific characteristics associated with low sport participation among UK adolescents, and (2) assess gender-specific BMI tracking, and gender-specific associations between BMI and self-esteem based on different levels of adolescent sport participation. Compared to Q4 boys and girls, Q1 boys and girls were more likely to be non-White, low family income, have overweight/obesity at 11 years and report lower self-esteem at 11 years and 14 years. Regardless of sport participation level, BMI tracks strongly among boys and girls between 11 and 14 years. We found that gender and sport participation influence the association between BMI and self-esteem. BMI at 11 years was inversely associated with self-esteem at 11 years among Q1 and Q2 boys and Q1 and Q4 girls. Moreover, BMI at 11 years was inversely associated with self-esteem scores at 14 years for Q1, Q3 and Q4 boys, and Q1, Q2, Q3 and Q4 girls.

For both boys and girls, low sport participation at 11 years was characterised by non-White ethnicity, low family income, overweight/obesity and low self-esteem. Building on the findings of a recent review [36], our findings highlight the need for broader intervention approaches to address inequalities in adolescent sport participation. In the present study, sport participation inequalities were more profound among girls than boys suggesting that girls may benefit most from interventions aimed at improving access and opportunity to adolescent sport. A current research and practice challenge is to understand how best to promote and support sport participation among adolescents least likely to engage in physical activity, such as low-income adolescents and adolescents with overweight/obesity and low self-esteem. There are known financial, geographical and transportation barriers associated with adolescent sport participation [39]. Concessionary schemes in UK leisure facilities are an effective way to increase physical activity levels and reduce inequalities [40,41], and could be an effective public health strategy to promote and support sport participation among UK adolescents and reduce health inequalities. Although, capability and motivation are also important facets of behaviour change and would need to be considered alongside improving opportunity [42].

In the present study, BMI tracking was examined over a 3-year period between 11 and 14 years. We revealed that regardless of sport participation level, BMI tracks strongly among boys and girls between early adolescence and mid-adolescence. This finding supports and builds on evidence linking childhood overweight with adolescent overweight [3,4]. A UK study which assessed weight status over 7 years revealed that children with overweight at 7 years were almost 20 times more likely to have overweight at 14 years [4]. A more recent study that conducted gender stratified analyses revealed that female gender was a predictor of less favourable BMI trajectories [7]. By conducting gender stratified analyses based on sport participation levels we were able to elucidate potential reasons for adolescent differences in BMI tracking. For both boys and girls, BMI tracking was higher among the lowest sport participation group than the highest sport participation group which extends the evidence base in this area. These findings further highlight the importance of early prevention in adolescent overweight. Given that BMI tracking was strong among the high sport participation quartile underscores the need for health programmes that target the full range of adolescent energy balance behaviours including diet, sleep and screen time use [43,44].

Another novel aspect of the present study was the investigation of gender-specific associations between BMI and self-esteem based on different levels of sport participation. We found that BMI at 11 years was inversely associated with self-esteem at 11 years among Q1 and Q2 boys and Q1 and Q4 girls. These findings are consistent with evidence showing that adolescents who have overweight or a higher BMI report lower self-esteem than their healthier weight peers [20,21]. A Canadian longitudinal study found that adolescents with obesity at baseline had almost twice the odds of reporting low self-esteem four years later compared with children with health weight [22]. A more recent longitudinal study revealed that higher initial BMI scores were associated with slower decreases in appearance esteem over time after controlling for age, gender and parental education [45]. The present study is the first to evidence prospective associations by gender and sport participation quartile. Our findings suggest that gender and sport participation influence the prospective inverse association between BMI and self-esteem three years later. Interestingly, while the strength of association was consistent among boys, associations were stronger among the Q1 and Q2 quartiles than the Q3 and Q4 quartiles among girls.

A potential reason for these combined findings could be due to adolescents with overweight being more likely (than normal weight peers) to experience stigmatisation and/or teasing from their peers which can lead to social marginalisation and low self-esteem [46,47]. Some evidence suggests that girls experience more weight-related stigma than boys [47] which may explain why prospective associations between BMI and self-esteem across sport quartiles were stronger for girls than boys. Although correlations between BMI and weight stigma were consistent across genders in a recent review study [48]. It is however probable that adolescents with overweight that participate in sport infrequently experience greater peer relation challenges than their peers whom participate in sport regularly given that peer relation challenges and victimisation are linked with physical inactivity and weight status [49,50].

Given that boys and girls in the highest sport participation quartile were more likely to experience higher self-esteem at 11 years and 14 years compared to boys and girls in the lowest sport participation quartile further underscores the potential health benefits of sport for adolescents. Some previous longitudinal research has revealed positive associations between sport participation and later self-esteem [25,51]. For example, in a longitudinal study of females, team sport achievement experiences in early adolescence were positively associated with self-esteem three years later [52]. Sport participation affords adolescents opportunities to build sport competencies and, in turn, their self-concept of their abilities [51]. A recent longitudinal study found that sport participation in late childhood was positively associated with boys’ and girls’ perceived social competence four years later [53]. Sport participation also facilitates social interaction [23] which is among the strongest determinants of self-esteem across the lifespan [54]. During adolescence it is common for sport participation levels [55] and self-esteem to decline [9], and for psychological disorders to also arise [56,57]. As such, sport programmes at this age are vitally important as overall physical activity participation levels typically decline too [58]. However, additional formative research is needed with low active adolescents with overweight and/or low self-esteem to better understand their participation needs and how best to design future sport and physical activity programmes.

This study has several strengths. The study is the first to assess gender-specific BMI tracking, and gender-specific associations between BMI and self-esteem based on different levels of adolescent sport participation. Other strengths include its longitudinal design, and large heterogeneous sample which permitted gender stratified analyses. There are however some study limitations to note. Firstly, sport participation may have been subject to measurement error and social desirability from parents/carers. Additionally, our sport participation measure was based on frequency alone and was unable to determine the duration, intensity or type of sport participation (e.g., level of interdependence with teammates) which may influence associations with BMI and self-esteem, and warrants further investigation [59]. Moreover, sport participation was only assessed at 11 years and not 14 years which limited examination of sport participation trajectories and associations with BMI and self-esteem over time. This is another area of research that warrants further investigation.

## 5. Conclusions

Low sport participation among boys and girls was characterised by non-White ethnicity, low family income, overweight/obesity and low self-esteem. Boys and girls in the lowest sport participation quartile at 11 years were more likely to have overweight/obesity at 11 years and experience low self-esteem at 11 and 14 years compared to adolescents in the highest sport participation quartile. This study provides evidence that gender and sport participation influence BMI tracking and the BMI and self-esteem association among adolescents. The findings of this study further underscore the health benefits of sport for adolescents. Intervention programmes to promote and support adolescent sport participation and reduce inequalities are needed in the UK.

## Figures and Tables

**Table 1 ijerph-19-15579-t001:** Descriptive characteristics of sample.

Sport Quartile and Cut-Off Values	Boys Mean (SD) or %				Girls Mean (SD) or %			
**Variable**	Q1 <1 day/week (*n* = 963)	Q2 1 day/week (*n* = 848)	Q3 2 days/week (*n* = 987)	Q4 ≥3 days/week (*n* = 1725)	Q1 Never (*n* = 1078)	Q2 >never–1 day/week (*n* = 1162)	Q3 2 days/week (*n* = 908)	Q4 ≥3 days/week (*n* = 1375)
**Ethnicity**								
**White**	78.10	77.50	84.40	89.30	68.70	78.70	88.40	92.40
**Mixed**	2.40	2.50	2.60	2.80	3.30	3.40	2.20	2.20
**Indian**	4.40	3.70	2.30	2.10	3.60	3.40	2.00	1.30
**Pakistani and Bangladeshi**	10.60	10.30	5.50	2.40	18.20	7.50	4.20	1.70
**Black or Black British**	2.20	3.90	3.40	2.60	4.00	4.70	2.20	1.70
**Other Ethnic group**	2.40	2.20	1.70	0.90	2.10	2.30	1.00	0.70
**Family income quantile**								
**Lowest**	26.00	17.70	11.80	8.60	32.20	17.80	11.30	8.40
**Second**	20.80	19.30	18.40	12.60	24.30	22.70	16.00	11.50
**Third**	23.40	25.10	20.70	20.60	18.50	21.50	21.70	20.10
**Fourth**	16.60	22.10	25.20	27.40	14.70	21.10	25.30	27.10
**Highest**	13.30	15.80	23.90	30.80	10.40	16.90	25.70	32.90
**Age 11 BMI (kg/m^2^)**	19.29 (3.81)	19.03 (3.53)	18.84 (3.30)	18.66 (3.16)	19.28 (3.80)	19.46 (3.77)	19.30 (3.51)	19.00 (3.29)
**Weight status**								
**Overweight**	20.40	20.50	19.00	17.70	22.00	24.40	20.00	19.70
**Obesity**	8.40	6.70	5.50	3.20	7.00	6.50	6.20	4.70
**Age 14 BMI (kg/m^2^)**	21.30 (4.47)	20.97 (4.15)	20.75 (3.91)	20.57 (3.44)	21.93 (4.43)	22.06 (4.21)	21.95 (4.06)	21.69 (3.80)
**Weight status**								
**Overweight**	20.40	18.90	16.50	15.80	18.40	20.80	20.20	20.10
**Obesity**	9.30	8.00	7.10	4.50	8.90	8.20	6.90	4.90
**Age 11 self-esteem score**	16.85 (2.14)	16.91 (2.30)	17.07 (2.10)	17.33 (1.99)	16.77 (2.29)	16.87 (2.07)	16.83 (2.17)	17.09 (2.10)
**Age 14 self-esteem score**	15.98 (2.76)	16.24 (2.54)	16.31 (2.62)	16.69 (2.47)	14.56 (3.09)	14.76 (2.98)	14.84 (2.99)	15.20 (2.83)

BMI, body mass index; SD, standard deviation.

**Table 2 ijerph-19-15579-t002:** Gender-specific Q1 and Q4 sport participation quartile differences in dichotomous and continuous variables.

Sport Quartile	Boys		*p* Value	*d*	Girls		*p* Value	*d*
**Variable**	Q1 <1 day/week (*n* = 963)	Q4 ≥3 days/week (*n* = 1725)			Q1 Never (*n* = 1078)	Q4 ≥3 days/week (*n* = 1375)		
**Dichotomous**	% (95% CI)							
**Ethnicity** **Non-White**	21.90	10.70			31.30	7.60		
	OR = 2.34 (1.88 to 2.90)		<0.001		OR = 5.50 (4.34 to 6.97)		<0.001	
**Family income quantile** **Lowest**	26.00	8.60			32.20	8.40		
	OR = 3.71 (2.97 to 4.63)		<0.001		OR = 5.15 (4.10 to 6.48)		<0.001	
**Age 11 weight status** **Overweight/obese**	28.80	20.90			29.00	24.40		
	OR = 1.53 (1.27 to 1.83)		<0.001		OR = 1.26 (1.06 to 1.51)		0.01	
**Age 14 weight status** **Overweight/obese**	29.70	20.30			27.30	25.00		
	OR = 1.67 (1.39 to 2.00)		<0.001		OR = 1.12 (0.93 to 1.34)		0.22	
**Continuous**	Mean (SD)							
**Age 11 BMI (kg/m^2^)**	19.29 (3.81)	18.66 (3.16)	<0.001	0.2	19.28 (3.80)	19.00 (3.29)	0.06	0.1
**Age 14 BMI (kg/m^2^)**	21.30 (4.47)	20.57 (3.44)	<0.001	0.2	21.93 (4.43)	21.69 (3.80)	0.15	0.1
**Age 11 self-esteem score**	16.85 (2.14)	17.33 (1.99)	<0.001	0.2	16.77 (2.29)	17.09 (2.10)	<0.001	0.2
**Age 14 self-esteem score**	15.98 (2.76)	16.69 (2.47)	<0.001	0.3	14.56 (3.09)	15.20 (2.83)	<0.001	0.2

BMI, body mass index; CI, confidence interval; OR, odds ratio; SD, standard deviation; Q1 and Q4 represent lowest and highest sport participation quartiles, respectively.

## Data Availability

The datasets analysed during the current study are available from the UK Data Service: beta.ukdataservice.ac.uk/datacatalogue/series/series?id=2000031 (accessed on 26 October 2022).

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
