# Peer review of "The Influence of Adolescent Sport Participation on Body Mass Index Tracking and the Association between Body Mass Index and Self-Esteem over a Three-Year Period"

_ijerph, 2022, doi:10.3390/ijerph192315579_

Round 1
Reviewer 1 Report
This manuscript deals with the important topic of teenage and adolescent health related to physical activity. The objectives of the study are clearly set and the sample population of boys and girls is significant. However, the results of this comprehensive study are very poorly presented. The lesson is to present the descriptive and regressive analysis in addition to the tables with different graphs and pictures.
Line 09-26: The abstract should not start with the objectives and it should be structured differently, containing a brief description starting with the introduction, objectives, methodology, the most important results and conclusions
Line 57: It is not clearly explained why focus is on low sport participation among UK youth. What about gender- specific characteristics in a high sport participation group?
LINE 138-148: Results of Q2 and Q3 gruups of girls and boys did not mentioned and presented
Line 182: It should be noted which results are shown in Table 2.
Line 210: with what exactly shown results do you support this claim?
Line 247: Relate this statement to the results shown previously
Author Response
Manuscript number: ijerph-2025958
Title: The influence of sport participation on body mass index tracking and associations between body mass index and self-esteem among UK youth.
The author appreciates the time and efforts by the editor and referees in reviewing the manuscript. I have addressed all comments indicated in the review reports. Please find below the details of the changes made to the manuscript and responses to the reviewers’ comments. Two versions of the manuscript have been submitted, one version showing the changes made using ‘tracked changes’, and another ‘clean’ version. The page and line numbers below refer to the ‘tracked changes’ version of the manuscript.
Reviewer 1
This manuscript deals with the important topic of teenage and adolescent health related to physical activity. The objectives of the study are clearly set and the sample population of boys and girls is significant. However, the results of this comprehensive study are very poorly presented. The lesson is to present the descriptive and regressive analysis in addition to the tables with different graphs and pictures.
I am grateful for the reviewer’s positive feedback.
- Line 09-26: The abstract should not start with the objectives and it should be structured differently, containing a brief description starting with the introduction, objectives, methodology, the most important results and conclusions.
Thank you for raising this point. While this may be the standard structure used in some journals it is not consistent across all. The abstract structure used in present study is consistent with our previously published work in the International Journal of Environmental Research and Public Health (Fairclough et al. 2019; 2022; Noonan, 2018a; 2018b; Noonan et al. 2017). Inspiration was taken from these previous publications when structuring the abstract in the present study.
- Line 57: It is not clearly explained why focus is on low sport participation among UK youth. What about gender- specific characteristics in a high sport participation group?
Thank you for highlighting this point. Presently, the extent of inequalities in UK youth sport participation is unknown. This is an important line of enquiry given the international interest in enhancing youth sport participation and reducing participation inequalities. By way of assessing gender-specific Q1 and Q4 sport participation quartile differences in dichotomised (i.e., ethnicity, family income and weight status) and continuous variables (i.e., BMI and self-esteem), the present study was able to fill this evidence gap. Further justification for this study focus has been provided on Line 59 to Line 64 to enhance clarity for the reader.
- Line 138-148: Results of Q2 and Q3 groups of girls and boys did not mentioned and presented.
Thank you for highlighting this point. Keeping with the focus of study aim 1, only data for Q1 and Q4 sport participation quartiles were used. Additional detail has been provided in the analyses section of the manuscript to improve clarity and ensure that this aspect of the study is explicitly clear to the reader (Line 123 to Line 132).
- Line 182: It should be noted which results are shown in Table 2.
Thank you for highlighting this lack of clarity. A more explicit statement has now been included on Line 205 to Line 207 detailing the results presented in what is now Table 3. Table 3 presents adjusted regression analyses showing age 11 BMI as a predictor of age 14 BMI, age 11 self-esteem scores and age 14 self-esteem scores for each sport participation quartile by gender. Moreover, to enhance clarity, additional sub titles have been included on Line 209, Line 216 and Line 221 to reflect the specific analyses undertaken (Line 205 to Line 207).
- Line 210: with what exactly shown results do you support this claim?
Thank you for highlighting this error. The section has been revised to accurately reflect the results of the study and align with the content presented in other sections of the manuscript. The revised content is presented on Line 243 to Line 247.
- Line 247: Relate this statement to the results shown previously
Thank you for highlighting this point. The section has been revised to accurately reflect the results of the study and align with the content presented in other sections of the manuscript. The revised content is presented on Line 243 to Line 247 and Line 291 to Line 294.
References
Fairclough, S. J., Hurter, L., Dumuid, D., Gába, A., Rowlands, A. V., del Pozo Cruz, B., Cox, A., Crotti, M., Foweather, L., Graves, L. E. F., Jones, O., McCann, D., Noonan, R. J., Owen, M., Rudd, J., Taylor, S. L., Tyler, R., & Boddy, L. M. (2022). The physical behaviour intensity spectrum and body mass index in school-aged youth: a compositional analysis of pooled individual participant data. International Journal of Environmental Research and Public Health, 19, 8778.
Fairclough, S. J., Christian, D. L., Saint-Maurice, P. F., Hibbing, P., Noonan, R. J., Welk, G. J., Dixon, P., & Boddy, L. M. (2019). Calibration and validation of the Youth Activity Profile as a physical activity and sedentary behaviour surveillance tool for English youth. International Journal of Environmental Research and Public Health, 16, 3711.
Noonan, R. J. (2018a). Prevalence of childhood overweight and obesity in Liverpool between 2006 and 2012: evidence of widening socioeconomic inequalities. International Journal of Environmental Research and Public Health, 15, 2612.
Noonan, R. J. (2018b). Poverty, weight status and dietary intake among UK adolescents. International Journal of Environmental Research and Public Health, 15, 1224.
Noonan, R. J., Boddy, L. M., Knowles, Z. R., & Fairclough, S. J. (2017). Fitness, fatness and active school commuting among Liverpool Schoolchildren. International Journal of Environmental Research and Public Health, 14, 995.
Reviewer 2 Report
This was a very interesting paper with great strengths in its longitudinal design and large sample size. The manuscript was concise and well-written. I have a few comments and suggested revisions to consider prior to endorsing publication.
-Given the focus on gender differences in this study, I wanted to get some clarity on how this construct was measured and if sex-differences were also considered. Was gender measured using a binary parent-reported item? Along those lines, the introduction of existing gender-gaps in BMI, sport participation, and self-esteem - are these primarily due to the social construction of gender or biological basis of sex?
-Can you confirm if height and weight were parent-reported? If so, I'd perhaps list this as a limitation given the inherent risk of measurement error (possibly bias if parents of children with higher BMI are underestimating).
-were underweight children categorized as 'normal weight' in the analyses?
-can you provide more information on how the Rosenberg's Self Esteem Scale was shortened for its use in the present study? Has there been validity studies comparing the shortened with the full length? Please also describe the psychometric properties of the scale as used in children in this age range.
-How did you determine to categorize sport participation? Why not use it as a continuous variable?
-Line 105: Can you clarify whether Q2 and Q3 were removed from the analysis or if they were collapsed into Q1 and Q4, respectively?
-In the discussion, you've used the terms inequities and inequalities interchangeably - these are different concepts - I would suggest selecting which one is most appropriate for your point, and using it consistently
-In your discussion on lines 270-283, it may be worthwhile to consider including https://doi.org/10.1016/j.jadohealth.2019.09.017
as this study is quite similar in its examination of sport participation on social self concept over time in youth - they found an effect of sport over time, but no gender differences.
-A fairly minor point, but I found the use of the term 'overweight' as a disease term (i.e. "prevalence of adolescent overweight has increased") a little awkward, grammatically. I would suggest an edit to use the term as a characteristic or provide an explanation for why it is preferred to be used as is.
Author Response
Manuscript number: ijerph-2025958
Title: The influence of sport participation on body mass index tracking and associations between body mass index and self-esteem among UK youth.
The author appreciates the time and efforts by the editor and referees in reviewing the manuscript. I have addressed all comments indicated in the review reports. Please find below the details of the changes made to the manuscript and responses to the reviewers’ comments. Two versions of the manuscript have been submitted, one version showing the changes made using ‘tracked changes’, and another ‘clean’ version. The page and line numbers below refer to the ‘tracked changes’ version of the manuscript.
Reviewer 2
This was a very interesting paper with great strengths in its longitudinal design and large sample size. The manuscript was concise and well-written. I have a few comments and suggested revisions to consider prior to endorsing publication.
I am grateful for the reviewer’s positive feedback.
- Given the focus on gender differences in this study, I wanted to get some clarity on how this construct was measured and if sex-differences were also considered. Was gender measured using a binary parent-reported item? Along those lines, the introduction of existing gender-gaps in BMI, sport participation, and self-esteem - are these primarily due to the social construction of gender or biological basis of sex?
Thank you for raising an important point. Gender was measured using a binary parent-reported item (i.e., male or female). The binary category has now been included in the respective section of the manuscript on Line 113. Sex differences [in addition to gender differences] were not considered as they were beyond the scope of the present study. The decision to focus on gender rather than sex was informed by the data available, in addition to recent research using similar datasets to the present study that focused solely on gender (Bannink, Pearce & Hope, 2016; Kelly et al. 2016a; 2016b; 2018; Patalay & Fitzsimons, 2021; Patalay & Gage, 2019; Zilanawala, Sacker & Kelly, 2017). Including previous published work (Noonan & Fairclough, 2018).
- Can you confirm if height and weight were parent-reported? If so, I'd perhaps list this as a limitation given the inherent risk of measurement error (possibly bias if parents of children with higher BMI are underestimating).
Thank you for highlighting this point. Participant height and weight was measured objectively at both ages. This point has now been explicitly stated in the relevant sections of the manuscript on Line 13 and Line 88.
- Were underweight children categorized as 'normal weight' in the analyses?
The normal weight category included underweight participants due to the dataset available. This detail has now been explicitly stated on Line 90 to improve clarity for the reader.
- Can you provide more information on how the Rosenberg's Self Esteem Scale was shortened for its use in the present study? Has there been validity studies comparing the shortened with the full length? Please also describe the psychometric properties of the scale as used in children in this age range.
The present study used the shortened version of Rosenberg’s Self-Esteem Scale as it was the self-esteem measure available in the dataset. The shortened version comprises five items reflecting a positive view of self. Rather than ten items (i.e., full version) reflecting both a positive and negative view of self. Item responses are reported on a 4-point scale ranging from strongly disagree to strongly agree. Responses were summed to provide a score ranging from 5 to 20, with higher scores indicating greater self-esteem. The alpha coefficient inter-item reliability was 0.74. The shortened version of Rosenberg’s Self-Esteem Scale measure has been used in several previous studies involving youth (e.g., Atkin et al. 2021; Bannink, Pearce & Hope, 2016; Kelly et al. 2016a; 2018; Mueller & Flouri, 2021; Zilanawala, Sacker & Kelly, 2017).
- How did you determine to categorize sport participation? Why not use it as a continuous variable?
Thank you for highlighting this point. The second aim of the study was to assess gender-specific BMI tracking, and gender-specific associations between BMI and self-esteem based on different levels of youth sport participation. Categorising sport participation in this way - based on gender-specific quartiles ensured that sample sizes were relatively even across categories (compared to the continuous variable; seven categories). The approach was informed by previous research including our own work which adopted a similar approach to categorise physical activity levels (Fairclough et al. 2015; Noonan & Fairclough, 2018). The analyses conducted in the present study were informed by this evidence.
- Line 105: Can you clarify whether Q2 and Q3 were removed from the analysis or if they were collapsed into Q1 and Q4, respectively?
Thank you for highlighting this point. Keeping with the focus of study aim 1, only data for Q1 and Q4 sport participation quartiles were used. Additional detail has been provided in the analyses section of the manuscript to improve clarity for the reader (Line 123 to Line 132).
- In the discussion, you've used the terms inequities and inequalities interchangeably - these are different concepts - I would suggest selecting which one is most appropriate for your point, and using it consistently
Thank you for highlighting this oversight. The term “inequity” has been replaced with “inequality” to improve consistency throughout the manuscript (Line 251).
- In your discussion on lines 270-283, it may be worthwhile to consider including https://doi.org/10.1016/j.jadohealth.2019.09.017. As this study is quite similar in its examination of sport participation on social self-concept over time in youth - they found an effect of sport over time, but no gender differences.
Thank you for sharing this information. The work of Bedard and colleagues (2020) has been included in the relevant section of the manuscript on Line 316 to Line 318 and the relevant citation has been included in the reference list on Line 476 to Line 477.
- A fairly minor point, but I found the use of the term 'overweight' as a disease term (i.e. "prevalence of adolescent overweight has increased") a little awkward, grammatically. I would suggest an edit to use the term as a characteristic or provide an explanation for why it is preferred to be used as is.
Thank you for highlighting this point. The decision to present the content in this way was informed by prior studies in the area (Bygdell et al. 2021; Noonan, 2018; Sanyaolu et al. 2019; Wang et al. 2020) as well as World Health Organization publications (World Health Organization 2021). However, the sentence has been revised slightly to improve clarity (Line 32).
References
Atkin, A. J., Dainty, J. R., Dumuid, D., Kontostoli, E., Shepstone, L., Tyler, R., Noonan, R. J., Richardson, C., & Fairclough, S. J. (2021). Adolescent time-use and mental health: A cross-sectional, compositional analysis in the Millennium Cohort Study. BMJ Open, 11, e047189.
Bannink, R., Pearce, A., & Hope, S. (2016). Family income and young adolescents’ perceived social position: associations with self-esteem and life satisfaction in the UK Millennium Cohort Study. Archives of Disease in Childhood, 101, 917–921.
Bedard, C., Hanna, S., & Cairney, J. (2020). A Longitudinal Study of Sport Participation and Perceived Social Competence in Youth. Journal of Adolescent Health 66 (2020) 352-359.
Bygdell, M., Célind, J., Lilja, L., Martikainen, J., Simonson, L., Sjögren, L., Ohlsson, C., & Kindblom, J. M. (2021). Prevalence of overweight and obesity from 5 to 19 years of age in Gothenburg, Sweden. Acta Paediatrica, 110, 3349–3355.
Fairclough, S. J., Boddy, L. M., Mackintosh, K. A., Valencia-Peris, A., & Ramirez-Rico, E. (2015). Weekday and weekend sedentary time and physical activity in differentially active children. Journal of Science and Medicine in Sport, 18, 444–449.
Kelly, Y., Patalay, P., Montgomery, S., & Sacker, A. (2016a). BMI development and early adolescent psychosocial well-being: UK Millennium Cohort Study. Pediatrics 138:967
Kelly, Y., Britton, A., Cable, N., Sacker, A., & Watt, R. G. (2016b). Drunkenness and heavy drinking among 11-year olds - Findings from the UK Millennium Cohort Study. Preventive Medicine 90, 139–142.
Kelly, Y., Zilanawala, A., Booker, C., & Sacker, A. (2018). Social Media Use and Adolescent Mental Health: Findings From the UK Millennium Cohort Study. EClinicalMedicine 6, 59–68.
Mueller, M. A. E., & Flouri, E. (2021). Urban Adolescence: The Role of Neighbourhood Greenspace in Mental Well-Being. Frontiers in Psychology, 12, 712065.
Noonan, R. J. (2018). Prevalence of childhood overweight and obesity in Liverpool between 2006 and 2012: Evidence of widening socioeconomic inequalities. International Journal of Environmental Research and Public Health, 15, 2612.
Noonan, R. J., & Fairclough, S. J. (2018). Cross-sectional associations between body mass index and social-emotional wellbeing among differentially active children. European Journal of Public Health, 29(2), 303–307.
Patalay, P., & Gage, S. H. (2019). Changes in millennial adolescent mental health and health-related behaviours over 10 years: a population cohort comparison study. International Journal of Epidemiology, 0(0), 1–15.
Patalay, P., & Fitzsimons, E. (2021). Psychological distress, self-harm and attempted suicide in UK 17-year olds: prevalence and sociodemographic inequalities. British Journal of Psychiatry, 219(2), 437-439.
Sanyaolu, A., Okorie, C., Qi, X., Locke, J., & Rehman, S. (2019). Childhood and Adolescent Obesity in the United States: A Public Health Concern. Global Pediatric Health, 6, 2333794X19891305.
Wang, Y., Beydoun, M. A., Min, J., Xue, H., Kaminsky, L. A., & Cheskin, L. J. (2020). Has the prevalence of overweight, obesity and central obesity levelled off in the United States? Trends, patterns, disparities, and future projections for the obesity epidemic. International Journal of Epidemiology, 49(3), 810–823.
World Health Organization. (2021). Obesity and overweight. Available at: https://www.who.int/news-room/fact-sheets/detail/obesity-and-overweight
Zilanawala, A., Sacker, A., & Kelly, Y. (2017). Longitudinal latent cognitive profiles and psychosocial well-being in early adolescence. Journal of Adolescent Health, 61(4), 493–500.
Reviewer 3 Report
The research topic is very interesting and current. The manuscript is methodologically very well done. Additional value to the manuscript is given by the longitudinal study and a large number of respondents. There are some minor mistakes that need to be fixed:
1. Self-citation should be avoided, and the author of the manuscript has 3 self-citations
2. Line 39 - this sentence is missing a quote
3. Line 87 - kg m-2 should be corrected in kg/m2
4. Lines 100-106 - it should be explained how the data on the participation of 14-year-olds in sports was obtained
5. Line 122 - a sentence cannot begin with a test mark χ2
6. Lines 134-135 - this data is not seen in Table 1?
7. Lines 138-148 - why is there no overview of this data through the Table?
8. Line 190 - 11 should be corrected in 14
Author Response
Manuscript number: ijerph-2025958
Title: The influence of sport participation on body mass index tracking and associations between body mass index and self-esteem among UK youth.
The author appreciates the time and efforts by the editor and referees in reviewing the manuscript. I have addressed all comments indicated in the review reports. Please find below the details of the changes made to the manuscript and responses to the reviewers’ comments. Two versions of the manuscript have been submitted, one version showing the changes made using ‘tracked changes’, and another ‘clean’ version. The page and line numbers below refer to the ‘tracked changes’ version of the manuscript.
Reviewer 3
The research topic is very interesting and current. The manuscript is methodologically very well done. Additional value to the manuscript is given by the longitudinal study and a large number of respondents. There are some minor mistakes that need to be fixed:
I am grateful for the reviewer’s positive feedback.
- Self-citation should be avoided, and the author of the manuscript has 3 self-citations
Thank you for raising this point. It is not uncommon for authors to cite previous works. The present study not only uses similar datasets to the present study but also builds on the findings of these three self-cited studies. As such, the aforementioned studies are very closely aligned to the present study in terms of sample characteristics and research design. The rationale for using these self-citations and not others was driven by these factors.
- Line 39 - this sentence is missing a quote
A quotation has not been included on Line 40 as the statement is simply describing the key term (i.e., self-esteem).
- Line 87 - kg m-2 should be corrected in kg/m2
Thank you for highlighting this error. This error has now been revised on Line 89.
- Lines 100-106 - it should be explained how the data on the participation of 14-year-olds in sports was obtained.
Sport participation data was parent reported at age 11 years only. This information is explicitly stated on Line 103. Sport participation data was not available in the dataset for age 14 and was beyond the scope of the present study. It is stated in the limitations section of the manuscript on Line 335 to Line 337, that this was a limitation of the present study.
- Line 122 - a sentence cannot begin with a test mark χ2
Thank you for highlighting this error. This particular section of the manuscript has been revised to ensure that chi-square is stated in full. The revised content is presented on Line 132.
- Lines 134-135 - this data is not seen in Table 1?
Thank you for highlighting this oversight. On reflection this descriptive information adds little to the section. As such, it has been removed. The sub section titles have been revised to reflect the revisions made to the section. The revised content is presented on Line 146 to Line 152.
- Lines 138-148 - why is there no overview of this data through the Table?
Thank you for highlighting this oversight. An additional table (i.e., Table 2) has been included in the results section detailing the gender-specific Q1 and Q4 sport participation quartile differences in dichotomous and continuous variables (i.e., study aim one). The revised content is presented on Line 197 to Line 200.
- Line 190 - 11 should be corrected in 14.
Thank you for raising this point. To improve clarity for the reader addition sub titles have been included on Line 209, Line 216 and Line 221 as well as in the title of Table 2 (Line 228 to Line 230).
Round 2
Reviewer 1 Report
The manuscript has been improved to a certain extent on the basis of reviews and is a result of interest to the wider social community.
Author Response
Manuscript number: ijerph-2025958
Title: The influence of sport participation on body mass index tracking and associations between body mass index and self-esteem among UK youth.
The author appreciates the time and efforts by the editor and referees in reviewing the manuscript.
Reviewer 1
The manuscript has been improved to a certain extent on the basis of reviews and is a result of interest to the wider social community.
I am grateful for the reviewer’s positive feedback.